# Design, Synthesis, and Anti-Cervical Cancer and Reversal of Tumor Multidrug Resistance Activity of Novel Nitrogen-Containing Heterocyclic Chalcone Derivatives

**DOI:** 10.3390/molecules28114537

**Published:** 2023-06-03

**Authors:** Zheng Yang, Zheng-Ye Liu, Mourboul Ablise, Aikebaier Maimaiti, Zuohelaguli Mutalipu, Yusupuwajimu Alimujiang, Aizitiaili Aihaiti

**Affiliations:** College of Pharmacy, Xinjiang Medical University, Urumqi 830011, China; qhyzcr@163.com (Z.Y.); lzy17599832619@163.com (Z.-Y.L.); akpar15@163.com (A.M.); zuohela@163.com (Z.M.); 13209922075@163.com (Y.A.); m17590823808@163.com (A.A.)

**Keywords:** azacyclic, glycyrrhiza chalcone, anti-cervical cancer activity, cisplatin resistance, molecular docking

## Abstract

This study involved the design and synthesis of 21 new nitrogen-containing heterocyclic chalcone derivatives utilizing the active substructure splicing principle, with glycyrrhiza chalcone serving as the lead compound. The targets of these derivatives were VEGFR-2 and P-gp, and their efficacy against cervical cancer was evaluated. Following preliminary conformational analysis, compound **6f** ((E)-1-(2-hydroxy-5-((4-hydroxypiperidin-1-yl)methyl)-4-methoxyphenyl)-3-(4-((4-methylpiperidin-1-yl)methyl)phenyl)prop-2-en-1-one) exhibited significant antiproliferative activity against human cervical cancer cells (HeLa and SiHa) with IC_50_ values of 6.52 ± 0.42 and 7.88 ± 0.52 μM, respectively, when compared to other compounds and positive control drugs. Additionally, this compound demonstrated lower toxicity towards human normal cervical epithelial cells (H8). Subsequent investigations have demonstrated that **6f** exerts an inhibitory impact on VEGFR-2, as evidenced by its ability to impede the phosphorylation of p-VEGFR-2, p-PI3K, and p-Akt proteins in HeLa cells. This, in turn, results in the suppression of cell proliferation and the induction of both early and late apoptosis in a concentration-dependent manner. Furthermore, **6f** significantly curtails the invasion and migration of HeLa cells. In addition, **6f** had an IC_50_ of 7.74 ± 0.36 μM against human cervical cancer cisplatin-resistant HeLa/DDP cells and a resistance index (RI) of 1.19, compared to 7.36 for cisplatin HeLa cells. The combination of **6f** and cisplatin resulted in a significant reduction in cisplatin resistance in HeLa/DDP cells. Molecular docking analyses revealed that **6f** exhibited binding free energies of −9.074 and −9.823 kcal·mol^−1^ to VEGFR-2 and P-gp targets, respectively, and formed hydrogen bonding forces. These findings suggest that **6f** has potential as an anti-cervical cancer agent and may reverse cisplatin-resistant activity in cervical cancer. The introduction of the 4-hydroxy piperidine and 4-methyl piperidine rings may contribute to its efficacy, and its mechanism of action may involve dual inhibition of VEGFR-2 and P-gp targets.

## 1. Introduction

Chalcone, a natural product with an α, β-unsaturated carbonyl system containing three carbon atoms and a structural parent of 1,3-diphenyl-1,2-propen-1-one composed of two aromatic rings, is considered an essential active compound [1] (Figure 1, chalcone). This compound is commonly found in medicinal plants, including licorice and safflower, and has demonstrated antifungal, anti-inflammatory, and antitumor biological activities [2]. Its straightforward synthesis, multiple reaction centers, and versatility in combining with various reactive groups have garnered significant attention in the scientific community [3]. Numerous chalcone compounds have demonstrated remarkable biological activity in the field of antitumor research. Through various screening experiments, chalcone has exhibited the ability to target a diverse array of cellular molecules. Representative mechanisms of chalcones’ anticancer effects include MDM2/p53, tubulin, NF-kB, Wnt/β-catenin, Akt-FGF-2/TGF-β/VEGF, VEGF/VEGFR-2, HIF-1, MMP-2/9, and P-gp/MRP1/BCRP [4].

Cervical cancer ranks as the fourth most common cause of cancer-related fatalities among women worldwide, with 604,000 new cases (3.1%) and 342,000 deaths (3.4%) reported in 2020 [5]. Despite being the preferred chemotherapy agent for cervical cancer, cisplatin is associated with treatment failure due to its low selectivity, toxic side effects, tumor multidrug resistance, and susceptibility to recurrence [6].

The significant association between elevated expressions of vascular endothelial growth factor (VEGF) and vascular endothelial growth factor receptor 2 (VEGFR-2) in cervical precancerous lesions and invasive cancers with worsening and poorer clinicopathological parameters suggests that targeting the VEGF/VEGFR-2 signaling pathway is an effective strategy for treating cervical cancer [7,8]. The present study indicates that VEGFR-2 is significantly upregulated in various types of malignant solid tumors, such as cervical, ovarian, and breast cancers. Consequently, targeting the VEGFR-2 pathway could be a promising strategy to elicit potent antiangiogenic and antitumor effects in neoplastic growths [9,10].

Tumor multidrug resistance (MDR) is a significant contributor to tumor treatment failure and recurrence. The overexpression of ABCB1 (P-gp, also known as P-glycoprotein) by tumor cells is a definitive cause of MDR [11]. The upregulation of P-gp has been widely demonstrated to cause multidrug resistance in cervical cancer, while high P-gp expression in tumor cells is responsible for cisplatin resistance [12]. Consequently, there is a pressing need to address the issue of cisplatin resistance while simultaneously mitigating the toxicity of cisplatin chemotherapy.

The inhibition of the VEGF/VEGFR-2 signaling pathway has been found to impede angiogenesis in tumor cells. Several derivatives, including hydroxy- and methoxy chalcones **1** [13], **2** [14], and **3** [15], as well as azetidine-containing chalcone derivatives **4** [16] and **5** [17], have demonstrated potential inhibitory activity against VEGFR-2 kinase. This kinase is closely linked to tumor cell proliferation and angiogenesis. The modulation of ABCG2/P-gp/BCRP by chalcone is a crucial factor in intracellular drug accumulation and resistance to conventional chemotherapeutic agents [18,19]. Research has demonstrated that chalcone derivative **6**, when combined with paclitaxel (DOX), competitively inhibits the P-gp drug efflux pump, leading to increased DOX concentration in resistant cells and enhanced antitumor activity against resistant cells [20] (Figure 1). The replacement of the urea group in sorafenib with the α,β-unsaturated aryl ketone of chalcone has shown potential for enhancing its cytotoxic activity. Specifically, compound **8** exhibited superior cytotoxic effects on HepG2, MCF-7, and PC-3 cells, with IC_50_ values of 0.56 ± 0.83, 3.88 ± 1.03, and 3.15 ± 0.81 μM, respectively. These values were 1.03–6.14 times higher than those of sorafenib, indicating improved antitumor activity. Furthermore, compounds **7**, **8**, and **9** demonstrated IC_50_ values of 0.72, 3.2, and 1.1 μM, respectively, against VEGFR-2 [21] (Figure 2).

Despite the fact that natural chalcones exhibit some anticancer activity and possess a straightforward synthesis process due to their simple structural composition, their drug-generating potential is restricted by several drawbacks, including inadequate water solubility, low bioavailability, unstable thermodynamic properties, and limited cell selectivity. Nitrogen-containing heterocycles are crucial pharmacodynamic entities in the design of antitumor drugs, and the majority of novel molecularly targeted antitumor drugs incorporate modifications of nitrogen heterocycles in their chemical structures [22]. The incorporation of nitrogen-containing heterocycles has been shown to augment the biological activity and hydrophilicity of the parent compound [23,24]. Notably, a prevalent characteristic of numerous MDR regulators is the existence of at least one elemental nitrogen [20].

Piperazine, a crucial pharmaceutical intermediate, exhibits favorable biological activities and low toxicity. The nitrogen atom present on its ring plays a significant role in the formation of hydrogen bonds within living organisms, thereby enhancing the affinity and selectivity of drugs towards receptors. Consequently, this class of compounds holds immense importance in the realm of drug development [25,26]. Pyrrolidine side chains are observed in the molecular structures of certain bioactive molecules and marketed drugs due to their exceptional activity as molecular structure fragments [27]. In antitumor drug research, piperidine side chains and morpholine ring derivatives are considered valuable pharmacodynamic groups [28,29]. Recently, novel piperazine–chalcone hybrids and related pyrazoline derivatives have been synthesized and designed as potential inhibitors of VEGFR-2 [17]. It is noteworthy that a significant number of MDR modulators possess at least one elemental nitrogen, with piperazine groups being more prevalent [20]. Hence, the hybridization of chalcone parent nuclei with other nitrogenous heterocyclic anticancer moieties to yield aza chalcone derivatives presents a promising avenue for enhancing anticancer activities and surmounting tumor resistance [4].

The failure of chemotherapy in cervical cancer is primarily attributed to chemotherapy-associated toxicity and tumor multidrug resistance. To address this challenge, research endeavors have been directed towards the development of novel, efficacious, and low-toxicity antitumor drugs that target potential therapeutic targets in cancer cell signaling pathways and employ structural optimization pathways of natural lead compounds [30]. Previous research has demonstrated that chalcone derivatives possess notable anti-cervical cancer properties while exhibiting lower toxicity towards normal cells. Additionally, the anti-cervical cancer effects of glycyrrhiza chalcone derivatives were found to be considerably heightened through methylation and halogenation modifications, utilizing glycyrrhiza chalcone as the primary compound [31] (Figure 3). The inhibitory effects of chalcone derivatives on VEGFR-2 or P-gp activities suggest that the chalcone parent nucleoskeleton structure holds significant potential for further research in antitumor drug development. Nitrogen-containing heterocycles, which are crucial pharmacodynamic groups in antitumor drug design and commonly found in VEGFR-2 and P-gp inhibitors, play a vital role in this regard.

Here, we present the design and synthesis of a novel nitrogen-containing heterocyclic chalcone derivative, **6f** ((E)-1-(2-hydroxy-5-((4-hydroxypiperidin-1-yl)methyl)-4-methoxyphenyl)-3-(4-((4-methylpiperidin-1-yl)methyl)phenyl)prop-2-en-1-one, which exhibits significant potential as an antitumor agent. Furthermore, we examine the fundamental mechanism underlying its anti-cervical cancer and tumor resistance reversal properties.

## 2. Results and Discussion

### 2.1. Chemistry

Figure 1, Figure 2, Figure 3 and Figure 4 demonstrate the synthesis of nitrogen-containing heterocyclic chalcone derivatives (**6a**–**6u**). Resorcinol was utilized as the starting material to obtain intermediate **1** using nucleophilic substitution under the reaction conditions of zinc chloride and glacial acetic acid. Intermediate **2** was produced by protecting intermediate **1** with phenolic hydroxyl groups in potassium carbonate and acetone. Intermediate **2** underwent Blanc chloromethylation in the presence of glacial acetic acid and chloromethyl methyl ether to yield intermediate **3** (Figure 1).

Intermediates **4a**–**4f** were synthesized via the reaction of intermediate **3** with 4-hydroxy piperidine, 4-methyl piperazine, 4-methoxy piperidine, 4-methyl piperidine, morpholine ring, and pyrrolidine in dichloromethane, utilizing triethylamine as a catalyst (Figure 2).

Intermediates **5a**–**5f** were subsequently prepared by reacting chloromethyl methyl ether with the aforementioned six nitrogen-containing heterocycles in dichloromethane or THF, also utilizing triethylamine as a catalyst. The purification process employed for intermediates **4a**–**4f** was also utilized for intermediates **5a**–**5f** (Figure 3).

Chalcone derivatives **6a**–**6u** were prepared using the Claisen–Schmidt reaction principle, using intermediates **4a**–**4f** and **5a**–**5f** as raw materials, dissolved in anhydrous ethanol and 20% KOH solution as the catalyst at a reaction temperature of 50 °C. The compounds were yellow oily substances with yields ranging from 31.59% to 79.18%, of which compound **6t** had the highest yield. ^1^H-NMR, ^13^C-NMR, and HRMS performed structural characterization and molecular weight confirmation of the intermediates and chalcone derivatives (Figure 4).

### 2.2. In Vitro Activity Assays

#### 2.2.1. Anti-Cervical Cancer Activity Assay

The MTT assay results indicate that compounds **6a**–**6u** possess proliferation inhibitory activity against HeLa, SiHa, and H8 cells, with compound **6f** exhibiting noteworthy anti-cervical cancer activity. In comparison to cisplatin and sorafenib, compound **6f** demonstrated significantly stronger antiproliferative effects on HeLa and SiHa cells (*p* < 0.05), while exhibiting lower toxicity towards human cervical normal H8 cells (Table 1).

The antiproliferative activities of **6f** and **6k** against HeLa and SiHa cells were found to be significantly greater than those of the lead compound, chalcone (*p* < 0.05). This suggests that the anti-cervical cancer activities of chalcone were notably improved through its coupling with the nitrogen-containing heterocycle. The impact of various nitrogen-containing substituted heterocycles on the anti-cervical cancer activity of chalcone derivatives will be examined in relation to conformational relationships. The VEGFR-2 inhibitor, sorafenib, demonstrated favorable antiproliferative effects on cervical cancer cells, as evidenced (Table 1). Further research is warranted to determine if the anti-cervical cancer properties of compounds **6f** and **6k** are associated with VEGFR-2 expressions.

#### 2.2.2. Structure–Effect Relationship Analysis

The antiproliferative activities of compounds **6a**–**6u** varied among two types of cervical cancer cells (HeLa and SiHa), with the most notable effects observed on HeLa cells and minimal toxicity to normal cervical H8 cells. A brief summary of the conformational relationship between the active effects of **6a**–**6u** on HeLa cells is presented below.

Compounds **6a**–**6f** had 4-hydroxy piperidine ring substitution on the A-ring, and when the B-ring was substituted using different nitrogen-containing heterocycles, the trend of the antiproliferative activities was: 4-methyl piperidine > pyrrolidine > 4-methoxy piperidine > morpholine ring > 4-hydroxy piperidine > 4-methyl piperazine, where the **6f** anticancer activities of the B-ring substituted using 4-methyl piperidine were more significant than those of the other compounds. The A-rings for **6g** and **6h** were substituted using 4-methoxy piperidine and the activity strengths for the B-ring substituent groups were 4-methyl piperidine > morpholine ring; the A-rings for **6i**–**6m** were substituted with pyrrolidine and the activity strengths for the B-ring substituent groups were 4-methyl piperidine > morpholine ring > 4-hydroxy piperidine > 4-methyl piperazine > 4-methoxy piperidine. The A-rings for **6n**–**6p** were substituted with morpholine ring, and activities of the B-ring substituent group were 4-methyl piperidine > 4-methoxy piperidine > morpholine ring. The A-rings for **6q**–**6t** were substituted with 4-methyl piperidine, and activities of the B-ring substituent group were 4-methyl piperidine > 4-methyl piperazine > pyrrolidine > morpholine ring. There were highly significant antiproliferative effects on the two cervical cancer cells and less toxicity to normal cells when the A-ring for **6u** was 4-methyl piperazine and the B-ring was 4-methyl piperidine.

The preliminary structure–activity relationship analysis revealed that introduction of nitrogen-containing heterocycles into the chalcone skeleton significantly improved anti-cervical cancer activities, coupled with insignificant toxicity to normal cells. When the A-ring of **6a**–**6u** was replaced using different nitrogen-containing heterocycles, compounds with 4-methyl piperidine in ring B showed strong anti-cervical cancer activities, with those of **6f** and **6k** being significantly higher than those of other compounds. Particularly, the anti-cervical cancer activities of **6f** were significantly higher than those of cisplatin and sorafenib, with low toxicity levels. These findings imply that the 4-methyl piperidine ring has an important role in structural modification of chalcone mother nucleus and is a potential anticancer pharmacophore. Moreover, we established that derivative activities of chalcone substituted using the morpholine ring were poor. In the molecular docking study, it was found that the 4-methyl piperidine ring could form stable hydrophobic interactions with amino acid residues of the VEGFR-2 binding site, thus enhancing the stability of the bond between the compound and the target.

#### 2.2.3. In Vitro Anti-HUVEC Cell Activities

The findings indicate that compound **6f** exhibited an IC_50_ value of 7.14 ± 0.91 μM, which was comparable to sorafenib’s IC_50_ of 9.20 ± 1.22 μM, but significantly higher than the lead compound and **6k** (*p* < 0.05) (Table 2). The Western blot assay results demonstrated that sorafenib and **6f** significantly inhibited the phosphorylation of VEGFR-2 compared to the blank group with a significant difference (*p* < 0.05) (Figure 5). Compound **6f** exhibited a significant inhibition of VEGFR-2 phosphorylation at 4 and 6 μM compared to the control group (*p* < 0.05) and was more effective than the positive agent sorafenib.

The upregulation of VEGFR is known to facilitate various processes such as tumor cell proliferation, angiogenesis, adhesion, invasion, and metastasis, while simultaneously hindering the apoptosis of tumor cells [32]. The data indicate that compound **6f** exhibits a noteworthy inhibitory effect on the proliferation of HUVEC cells, which is comparable to that of sorafenib (Table 2). This finding suggests that the antiproliferative activity of **6f** on HUVEC cells may be attributed to the suppression of VEGFR-2 activity. The findings demonstrate that the administration of **6f** resulted in a dose-dependent inhibition of VEGFR-2 phosphorylation in comparison to the control group (Figure 5). This suggests that **6f** directly targeted the activation of VEGFR-2 in HUVEC cells, leading to the inhibition of angiogenesis. Angiogenesis plays a vital role in the development and metastasis of cancer, and in addition to conventional treatments such as chemotherapy and radiotherapy, inhibition of angiogenesis has emerged as a promising therapeutic strategy for malignancies [15]. Compound **6f** has been observed to inhibit the phosphorylation of VEGFR-2 in HUVEC cells, thereby potentially impacting tumor angiogenesis and offering the possibility of antitumor proliferation.

#### 2.2.4. In Vitro VEGFR-2 Inhibitory Assay

The IC_50_ value of **6f** on VEGFR-2 kinase was found to be 0.75 ± 0.05 μM, which was comparable to that of sorafenib (0.56 ± 0.04 μM) but significantly stronger than the lead compound chalcone (*p* < 0.05). In contrast, the IC_50_ value for compound **6k** was 1.67 ± 0.18 μM, which was significantly lower than those of sorafenib and **6f** (*p* < 0.05) (Table 3).

The VEGFR-2 kinase is an important target for developing potential anticancer drug candidates [33]. VEGFR-2 is currently highly expressed in a variety of malignant solid tumor cells, including cervical, ovarian, breast, lung, thyroid, intestinal, and melanoma, and inhibition of the VEGFR-2 pathway would lead to effective antiangiogenic and antitumor responses [9,10]. The present study employed human VEGFR-2 ELISA (enzyme-linked immunosorbent assay) and sorafenib as the reference drug to investigate the inhibitory effects of **6f** and **6k** on VEGFR-2. The results indicate that compound **6f** exhibited a significantly higher inhibitory effect on VEGFR-2 compared to the lead compound and **6k**, approaching that of the positive drug sorafenib. This observation suggests that the introduction of nitrogen-containing heterocycles, namely 4-hydroxy piperidine and 4-methoxy piperidine, to the structure of **6f** may be responsible for its enhanced inhibitory effect on VEGFR-2.

#### 2.2.5. Compound **6f** Blocked the PI3K/AKT Pathway of HeLa Cells

It was demonstrated that compound **6f** exhibited a concentration-dependent suppression of VEGFR-2 phosphorylation in HeLa cells. Treatment with 2, 4, and 6 μM of **6f** resulted in effective downregulation of p-VEGFR-2 (Y1175), p-PI3K (Tyr458), and p-AKT (Ser473). Notably, compound **6f** demonstrated a significantly higher inhibitory effect on VEGFR-2 phosphorylation than the control at 4 and 6 μM (*p* < 0.05). Additionally, the inhibitory effect of **6f** on PI3K phosphorylation was significantly higher than that of the control group (*p* < 0.05). At 6 μM, **6f** exhibited a significant ability to inhibit AKT phosphorylation (Figure 6).

The PI3K/Akt pathway, which serves as a prominent downstream signaling mechanism of VEGFR-2, plays a crucial role in the regulation of angiogenesis and vascular permeability. This is achieved through the enhancement of endothelial cell survival and proliferation, as well as the overactivation of this pathway in cancer cells [34]. The experimental findings indicate that compound **6f** may impede the proliferation of tumor cells by hindering the phosphorylation of VEGFR-2, thereby suppressing the expression of the PI3k/AKT signaling pathway and ultimately restraining the proliferation of tumor cells. Further experimentation is necessary to confirm whether the inhibition of the PI3K/AKT signaling pathway by compound **6f** induces apoptosis in HeLa cells.

#### 2.2.6. Compound **6f** Induced the Apoptosis of Hela Cells

The results demonstrate that compound **6f** exhibited proapoptotic effects on HeLa cells, with rates of 13.4%, 20.7%, and 34.6% observed after 24 h of treatment at concentrations of 2, 4, and 6 μM, respectively (Figure 7A,B). These rates were significantly higher than those observed in the control group (*p* < 0.05). Furthermore, the proapoptotic rate of **6f** at 6 μM was found to be greater than that of the positive control drug sorafenib at 31.7%. Western blot analysis revealed that **6f** significantly upregulated the expression of Bax at concentrations of 2, 4, and 6 μM compared to the control group (*p* < 0.05), while also significantly downregulating the expression of Bcl-2 (*p* < 0.05) (Figure 7C,D).

Apoptosis is a process that entails the activation, expression, and regulation of a sequence of internal pathway proteins, ultimately resulting in programmed cell death, which is crucial for maintaining tissue homeostasis [35]. To elucidate the mechanism of action of **6f** on HeLa cells, a flow cytometry-based annexin V-FITC/propidium iodide (PI) dual staining assay was conducted. The findings revealed that **6f** could induce both early and late apoptosis in HeLa cells in a concentration-dependent manner (Figure 7A,B). Moreover, the Western blot analysis elucidated that the administration of **6f** exhibited a dose-dependent increase in the expression of the proapoptotic protein Bax and a decrease in the expression of the antiapoptotic protein Bcl-2 (Figure 7C,D). In conjunction, the compound **6f** may impede the PI3K/Akt signaling pathway by hindering the phosphorylation of VEGFR-2, thereby downregulating Bcl-2 and upregulating the expression of Bax, culminating in the apoptosis of HeLa cells.

#### 2.2.7. Compound **6f** Inhibited the Migration and Invasion of HeLa Cells

It was observed that the compound **6f** exhibited a reduction in the penetration of HeLa cells through the chamber membrane with increasing concentrations over a 24 h period. In comparison to the control group, the number of migrating cells decreased significantly from 1205 to 372, 132, and 71, respectively (*p* < 0.05) (Figure 8A,B), and the number of invading cells decreased from 172 to 91, 50, and 35, respectively (*p* < 0.05) (Figure 8C,D).

Tumor growth is facilitated by the migration, invasion, and metastasis of tumor cells, wherein individual cells detach from the primary tumor and infiltrate the lymphatic vessels, blood, or other tissues [36]. The present study aimed to assess the invasion and migration potential of **6f** on HeLa cells using the transwell assay. Furthermore, the inhibitory effect of **6f** on HeLa cell invasion and migration was observed in a dose-dependent manner at a consistent concentration for 24 h. Notably, **6f** exhibited a significantly stronger impact on HeLa cell invasion and migration than the positive drug sorafenib at 4 μM. These findings suggest that the inhibitory effect of **6f** on VEGFR-2 may contribute to a reduction in tumor cell invasion and migration. It is worth noting that chemotherapy toxicity and tumor multidrug resistance remain the primary causes of treatment failure in cervical cancer. The noteworthy impact of compound **6f** on HeLa cells and its potential mechanism of action prompted an investigation into its efficacy against cisplatin-resistant cell lines in cervical cancer.

#### 2.2.8. In Vitro Anti-HeLa/DDP Cell Activities

The HeLa/DDP cell lines demonstrated a moderate level of resistance to cisplatin, paclitaxel, and doxorubicin, with corresponding resistance indices (RI) of 7.36, 6.21, and 3.93. The IC_50_ values for **6f** on both HeLa and HeLa/DDP cell lines were 6.52 ± 0.42 and 7.74 ± 0.36 μM (*p* < 0.05), respectively. However, the anti-HeLa/DDP cell activities of **6k** were significantly inferior to those of **6f** and sorafenib. Upon combining **6f** with cisplatin at concentrations of 0.25, 0.5, and 1 μM, the resulting reversal index (RI) values for cisplatin were reduced to 5.48, 3.04, and 1.44 (*p* < 0.05), respectively. These findings indicate a more pronounced reversal effect when compared to the RI value of 7.36 observed for cisplatin in isolation (*p* < 0.05) (Table 4).

The Western blot findings indicate that compound **6f** did not exhibit a significant variance in P-gp expression within the concentration range of 0.25, 0.5, and 1 μM (*p* > 0.05). Furthermore, **6f** demonstrated a noteworthy ability to impede the phosphorylation of VEGFR-2 in HeLa/DDP cells within the range of 4 and 6 μM when compared to the blank group (*p* < 0.05) (Figure 9A–D).

From a clinical perspective, MDR denotes the phenomenon of cross-resistance to antineoplastic drugs and other drugs of the same category that possess distinct structural types and mechanisms of action, following their prolonged administration during chemotherapy [37]. This research endeavor aimed to examine the inhibitory effects of four positive drugs, namely chalcone, **6f**, and **6k**, on the proliferation of HeLa/DDP cell lines. The results indicated that the inhibitory activities of **6f** were significantly superior to those of the other compounds, with an RI value of 1.19 and negligible drug resistance. Furthermore, the inhibitory rate of compound **6f** on the HeLa/DDP cell line within the concentration range of 0.25, 0.5, and 1 μM was less than 10%, and it did not manifest any anti-proliferative activities. The observed outcomes, which are likely attributable to the suppression of P-gp protein transport function on the surface of the HeLa/DDP cell membrane by **6f**, result in the augmented intracellular accumulation of cisplatin and the inhibition of HeLa/DDP cell proliferation by impeding the phosphorylation of VEGFR-2.

#### 2.2.9. Molecular Docking

Through the process of docking chalcone compounds **6f** and **6k** with crystal structures of VEGFR-2 (ID:4ASD) and P-gp (ID:7O9W), it was determined that the free binding energy of compound **6f** with VEGFR-2 and P-gp was −9.074 and −9.823 kcal·mol^−1^, respectively. This value was comparable to the binding energy of the positive drug sorafenib with VEGFR-2 (−11.403 kcal·mol^−1^), but greater than that of verapamil with P-gp (−7.507 kcal·mol^−1^) (Table 5). Furthermore, it was observed that compound **6f** established hydrogen bonding interactions with amino acid residues ALA-55 (2.7 Å) and ASN-149 (2.7 Å) of the VEGFR-2 and P-gp targets, respectively. The minimum free binding energies for compound **6k** with VEGFR-2 and P-gp were determined to be −8.646 and −8.826 kcal·mol^−1^, respectively. Compound **6k** was found to form hydrogen bonds with amino acid residues THR-96 (2.1 Å) and GLN-92 (2.7 Å) at the P-gp binding site and exhibited robust hydrophobic interactions with the neighboring amino acid residues (Figure 10A–H).

AutoDock Vina is a widely utilized open-source program for molecular docking known for its speed and efficiency [38]. Molecular docking analysis revealed that the nitrogen-containing heterocycles, specifically 4-methyl piperidine, 4-methoxy piperidine, and pyrrolidine, present in the structures of compounds **6f** and **6k**, exhibited strong hydrophobic interactions with amino acid residues located at the periphery of VEGFR-2 and P-gp, as well as the formation of hydrogen bonding forces. The enhanced stability of compound–protein binding resulting from this phenomenon may contribute to a more effective inhibitory impact. This could potentially elucidate the observed inhibitory effects of compounds **6f** and **6k** on cervical cancer progression, multidrug-resistant cervical cancer, and reversal of multidrug resistance in vitro in cervical cancer. Nevertheless, further investigation is necessary to fully comprehend the underlying mechanisms.

## 3. Material and Methods

### 3.1. Chemistry

All the solvents and reagents used in the current study were purchased from commercial suppliers and used without further purification. The melting points were measured using the WRX-4 micro melting point meter. Column chromatography silica gel particle size of 200 or 400 mesh was purchased from Qingdao Chemical Co. Thin layer chromatography (TLC) analysis was performed using silica gel 60 F254 analytical plates (Merck, Billerica, MA, USA). Reaction products were purified by crystallization or flash column chromatography using a mixture of ethyl acetate/methanol as the eluent. The ^1^H NMR and ^13^C NMR spectra were measured using a Unity-Inova600 spectrometer (Varian, Palo Alto, CA, USA) and Bruker Avance III 400 HD spectrometer (Bruker Bioscience, Billerica, MA, USA) with tetramethylsilane (TMS) as the internal standard. Chemical shifts (δ) were reported in parts per million downfield relative to tetramethylsilane. Thermo Scientific Q Exactive (Thermo Scientific, 81 Wyman Street, Waltham, MA, USA) mass spectrometer was also used. Using an Agilent 1220 high performance liquid chromatograph (Agilent Technologies Co., Ltd., Palo Alto, CA, USA), the purity of all tested compounds was ≥95% with a detection wavelength of 372 nm, mobile phase (MeOH:H_2_O = 70:30), and a flow rate of 1 mL/min.

### 3.2. Synthesis and Structural Characterization

#### 3.2.1. Synthesis of Intermediate **1** [39]

In a 100 mL three-necked flask containing 15 mL of ice-acetic acid, 11.10 g (100 mmol) of resorcinol and 7.8 g (57 mmol) of anhydrous ZnCl_2_ powder were added. The mixture was heated and stirred at 110–115 °C for 3 h. Upon cooling, a viscous liquid was obtained, which was then dripped into 30 mL of ice-distilled water using a pipette. Gradually, orange-red crystals were precipitated. The resulting mixture was filtered, washed with ice-distilled water, and dried at 50 °C. The final yield of orange-red solid was 9.2 g (60.47 mmol), representing a 60.47% yield. m.p. 143.5–144.5 °C; 1-(2,4-dihydroxyphenyl)ethan-1-one(**1**), ^1^H NMR (400 MHz, DMSO-*d*_6_) δ 12.65 (s, 1H), 10.57 (s, 1H), 7.65 (d, *J* = 8.8 Hz, 1H), 6.37 (dd, *J* = 8.8, 2.4 Hz, 1H), 6.28 (d, *J* = 2.4 Hz, 1H), 2.53 (s, 3H). ^13^C NMR (101 MHz, DMSO-*d*_6_) δ 202.93, 165.35, 164.81, 133.83, 113.23, 108.51, 102.85, 26.39. HRMS (ESI) *m*/*z* calcd for C_8_H_9_O_3_^+^ (M + H)^+^ 153.05462, found 153.05470.

#### 3.2.2. Synthesis of Intermediate **2** [40]

In this experiment, a 50.00 g (329 mmol) sample of intermediate **1** was introduced into a 500 mL round-bottom flask and dissolved in 150 mL of acetone. Subsequently, 67.62 g (490 mmol) of anhydrous potassium carbonate was added and stirred, followed by the addition of 66.76 g (530 mmol) of dimethyl sulfate through a drop funnel. The resulting mixture was stirred at 50 °C, leading to the formation of white solid precipitates. The reaction progress was monitored using thin-layer chromatography and terminated at the end of a 6 h period. The resulting brownish-red liquid was subjected to filtration, followed by extraction, evaporation of acetone, and crystallization of the remaining reaction mixture at 4 °C. The resulting orange flake solid, identified as intermediate **2**, was obtained in a yield of 65.05% after filtration and drying, with a mass of 35.60 g, m.p. 47.9–48.8 °C; 1-(2-hydroxy-4-methoxyphenyl)ethan-1-one(**2**), ^1^H NMR (400 MHz, chloroform-*d*) δ 12.74 (s, 1H), 7.61 (d, *J* = 8.8 Hz, 1H), 6.44 (d, *J* = 2.1 Hz, 1H), 6.41 (dd, *J* = 8.2, 2.2 Hz, 1H), 3.86 (s, 3H), 2.54 (s, 3H). ^13^C NMR (101 MHz, chloroform-*d*) δ 202.58, 166.10, 165.24, 132.31, 113.89, 107.57, 101.14, 100.83, 55.53, 26.17. HRMS (ESI) *m*/*z* calcd for C_9_H_11_O_3_^+^ (M + H)^+^ 167.07027, found 167.07037.

#### 3.2.3. Synthesis of Intermediate **3**

In a 100 mL round-bottom flask, 25.0 g (150 mmol) of intermediate **2** was introduced and dissolved in 50 mL of glacial acetic acid. Subsequently, 25.0 g (311 mmol) of chloromethyl methyl ether was added to the flask using a dropping funnel and stirred at room temperature for 3 h. The resulting mixture was precipitated to form a white solid and allowed to react for 6 h. The reaction mixture was then solidified, allowed to stand, filtered, and the filter cake was washed with anhydrous ice ethanol. The resulting product was dried at 50 °C to yield 15.99 g (74.5 mmol) of white needle crystal solid, which was identified as intermediate **3** with a yield of 49.67%, m.p. 115.4–116.3 °C; 1-(5-(chloromethyl)-2-hydroxy-4-methoxyphenyl)ethan-1-one(**3**), ^1^H NMR (400 MHz, chloroform-*d*) δ 12.84 (s, 1H), 7.68 (d, *J* = 1.2 Hz, 1H), 6.42 (s, 1H), 4.59 (d, *J* = 1.3 Hz, 2H), 3.91 (s, 3H), 2.57 (s, 3H). ^13^C NMR (101 MHz, chloroform-*d*) δ 202.54, 165.79, 163.77, 133.02, 117.73, 113.33, 99.72, 56.06, 41.34, 26.26. HRMS (ESI) *m*/*z* calcd for C_10_H_12_ClO_3_^+^ (M + H)^+^ 215.04695, found 215.04681.

#### 3.2.4. Synthesis of Intermediates **4a**–**4f**

In this preparation, 1.6 g (7.45 mmol) of intermediate **3** was introduced into a 100 mL three-neck flask, which was then supplemented with 25 mL of dichloromethane to facilitate dissolution. Subsequently, 1.51 g (14.90 mmol) of 4-hydroxy piperidine and 2 mL of triethylamine were added sequentially. The mixture was subjected to refluxing and stirring at 45 °C for a duration of 6 h until the solution darkened in color, white solids precipitated, and the reaction ceased. The solution’s pH was modified to a range of 2–3 through the utilization of dilute hydrochloric acid. Subsequently, the water layer was separated by the addition of potassium carbonate, and the pH was adjusted to a range of 9–10. The solution was then subjected to thrice extraction using ethyl acetate, followed by evaporation to eliminate the ethyl acetate. The resulting yellow oil was then solidified to yield 1.45 g (5.18 mmol) of intermediate **4a**. The synthesis procedures for intermediates **4b**–**4f** were identical to those employed for **4a**.

(1) *1-(2-hydroxy-5-((4-hydroxypiperidin-1-yl)methyl)-4-methoxyphenyl)ethan-1-one* (**4a**). A red-brown solid, yield 69.52%, m.p. 100.2–101.2 °C; ^1^H NMR (400 MHz, chloroform-*d*) δ 12.72 (s, 1H), 7.69 (s, 1H), 6.39 (s, 1H), 3.84 (s, 3H), 3.69 (td, *J* = 9.1, 4.7 Hz, 1H), 3.46 (s, 2H), 2.79 (dt, *J* = 10.7, 4.4 Hz, 2H), 2.57 (s, 3H), 2.54 (s, 1H), 2.25–2.14 (m, 2H), 1.89 (dt, *J* = 13.3, 4.1 Hz, 2H), 1.61 (ddt, *J* = 13.0, 9.5, 4.9 Hz, 2H). ^13^C NMR (101 MHz, chloroform-*d*) δ 202.88, 164.35, 164.29, 132.62, 118.09, 113.23, 99.02, 67.83, 55.73, 55.06, 50.94(2C), 34.49(2C), 26.28. HRMS (ESI) *m*/*z* calcd for C_15_H_22_NO_4_^+^ (M+H)^+^ 280.15433, found 280.15475.

(2) *1-(2-hydroxy-4-methoxy-5-((4-methylpiperazin-1-yl)methyl)phenyl)ethan-1-one* (**4b**). A white solid, yield 87.58%, m.p. 104.2–104.8 °C; ^1^H NMR (400 MHz, chloroform-*d*) δ 12.73 (s, 1H), 7.68 (s, 1H), 6.39 (s, 1H), 3.84 (s, 3H), 3.47 (s, 2H), 2.57 (s, 3H), 2.54–2.46 (m, 8H), 2.29 (s, 3H). ^13^C NMR (101 MHz, chloroform-*d*) δ 202.74, 164.37, 164.33, 132.43, 117.92, 113.20, 99.06, 55.71, 55.24, 55.20(2C), 52.98(2C), 46.05, 26.28. HRMS (ESI) *m*/*z* calcd for C_15_H_23_N_2_O_3_^+^ (M + H)^+^ 279.17032, found 279.17044.

(3) *1-(2-hydroxy-4-methoxy-5-((4-methoxypiperidin-1-yl)methyl)phenyl)ethan-1-one* (**4c**). A yellow oil, yield 69.15%; ^1^H NMR (400 MHz, chloroform-*d*) δ 12.69 (s, 1H), 7.67 (s, 1H), 6.34 (s, 1H), 3.82 (s, 3H), 3.42 (s, 2H), 3.31 (s, 3H), 3.19 (tq, *J* = 9.2, 5.0, 4.4 Hz, 1H), 2.75 (dt, *J* = 10.6, 4.5 Hz, 2H), 2.54 (s, 3H), 2.18 (ddd, *J* = 12.3, 9.8, 2.9 Hz, 2H), 1.90 (dq, *J* = 13.7, 3.8 Hz, 2H), 1.60 (ddd, *J* = 12.8, 8.5, 3.5 Hz, 2H). ^13^C NMR (101 MHz, chloroform-*d*) δ 202.49, 164.11, 164.07, 132.13, 118.12, 113.03, 98.76, 76.25, 55.47, 55.24, 54.98, 50.82(2C), 30.88(2C), 26.00. HRMS (ESI) *m*/*z* calcd for C_16_H_24_NO_4_^+^ (M+H)^+^ 294.16998, found 294.17044.

(4) *1-(2-hydroxy-4-methoxy-5-((4-methylpiperidin-1-yl)methyl)phenyl)ethan-1-one* (**4d**). A yellow solid, yield 59.75%, m.p. 64.9–65.7 °C; ^1^H NMR (400 MHz, chloroform-*d*) δ 12.71 (s, 1H), 7.68 (s, 1H), 6.39 (s, 1H), 3.84 (s, 3H), 3.43 (s, 2H), 2.88 (dt, *J* = 11.8, 3.4 Hz, 2H), 2.57 (s, 3H), 2.00 (td, *J* = 11.5, 2.4 Hz, 2H), 1.90 (td, *J* = 11.6, 2.5 Hz, 2H), 1.38–1.32 (m, 1H), 1.40–1.12 (m, 2H), 0.92 (d, *J* = 6.3 Hz, 3H). ^13^C NMR (101 MHz, chloroform-*d*) δ 202.83, 164.37, 164.21, 132.40, 118.51, 113.22, 98.98, 55.69, 55.66, 53.90(2C), 34.35(2C), 30.78, 26.31, 22.01. HRMS (ESI) *m*/*z* calcd for C_16_H_24_NO_3_^+^ (M + H)^+^ 278.17507, found 278.17459.

(5) *1-(2-hydroxy-4-methoxy-5-(morpholinomethyl)phenyl)ethan-1-one* (**4e**). A yellow solid, yield 72.19%, m.p. 92.4–93.6 °C; ^1^H NMR (400 MHz, chloroform-*d*) δ 12.73 (s, 1H), 7.68 (s, 1H), 6.40 (s, 1H), 3.85 (s, 3H), 3.72 (t, *J* = 4.7 Hz, 4H), 3.46 (s, 2H), 2.57 (s, 3H), 2.49 (t, *J* = 4.6 Hz, 4H). ^13^C NMR (101 MHz, chloroform-*d*) δ 202.72, 164.45, 164.41, 132.63, 117.51, 113.25, 99.16, 67.03(2C), 55.78, 55.77, 53.52(2C), 26.28. HRMS (ESI) *m*/*z* calcd for C_14_H_20_NO_4_^+^ (M + H)^+^ 266.13868, found 266.13864.

(6) *1-(2-hydroxy-4-methoxy-5-(pyrrolidin-1-ylmethyl)phenyl)ethan-1-one* (**4f**). A yellow solid, yield 91.14%, m.p. 71.3–72.4 °C; ^1^H NMR (400 MHz, chloroform-*d*) δ 12.74 (s, 1H), 7.66 (s, 1H), 6.39 (s, 1H), 3.85 (s, 3H), 3.57 (s, 2H), 2.57 (s, 3H), 2.56–2.49 (m, 4H), 1.80 (dt, *J* = 4.8, 4.1 Hz, 4H). ^13^C NMR (101 MHz, chloroform-*d*) δ 202.77, 164.32, 164.13, 132.41, 119.27, 113.19, 99.18, 55.75, 54.14(2C), 53.23, 26.32, 23.49(2C). HRMS (ESI) *m*/*z* calcd for C_14_H_20_NO_3_^+^ (M + H)^+^ 250.14377, found 250.14403.

#### 3.2.5. Preparation of Intermediates **5a**–**5f**

In this preparation, 1.0 g (6.47 mmol) of p-chloromethyl benzaldehyde was introduced into a 100 mL three-neck flask and supplemented with 20 mL of dichloromethane to facilitate dissolution. Subsequently, 1.2 g (11.86 mmol) of 4-hydroxypiperidine and 1 mL of triethylamine were sequentially added. The mixture was subjected to refluxing and stirring at 45 °C for a duration of 6 h until the reaction solution attained a dark and light yellow coloration. The reaction was terminated upon the formation of a white solid precipitate. The solution’s pH was subsequently modified to a range of 2–3 utilizing dilute HCl. Following this, potassium carbonate was added to the separation water layer to adjust the pH to a range of 9–10. The resulting precipitate was extracted using ethyl acetate, and the resulting light-yellow oil was evaporated to yield a solid paste weighing 0.82 g (3.74 mmol), identified as intermediate **5a**. The synthesis procedures for intermediates **5b**–**5f** were identical to those employed for **5a**.

(1) *4-((4-hydroxypiperidin-1-yl)methyl)benzaldehyde* (**5a**), a yellow solid, yield 57.78%, m.p. 64.9–65.6 °C; ^1^H NMR (400 MHz, DMSO-*d*_6_) δ 9.99 (s, 1H), 7.87 (d, *J* = 7.9 Hz, 2H), 7.52 (d, *J* = 7.9 Hz, 2H), 4.58 (d, *J* = 4.2 Hz, 1H), 3.53 (s, 2H), 3.48 (dt, *J* = 9.0, 4.6 Hz, 1H), 2.66 (dt, *J* = 10.4, 4.4 Hz, 2H), 2.06 (dd, *J* = 8.7, 6.0 Hz, 2H), 1.77–1.66 (m, 2H), 1.41 (dtd, *J* = 13.1, 9.6, 3.6 Hz, 2H). ^13^C NMR (101 MHz, DMSO-*d*_6_) δ 193.17, 146.67, 135.58, 129.95(2C), 129.63(2C), 66.62, 62.17, 51.39(2C), 34.85(2C). HRMS (ESI) *m*/*z* calcd for C_13_H_18_NO_2_^+^ (M + H)^+^ 220.13321, found 220.13322.

(2) *4-((4-methylpiperazin-1-yl)methyl)benzaldehyde* (**5b**), a yellow oil, yield 44.63%. ^1^H NMR (400 MHz, chloroform-*d*) δ 9.97 (s, 1H), 7.82 (d, *J* = 7.9 Hz, 2H), 7.50 (d, *J* = 7.9 Hz, 2H), 3.57 (s, 2H), 2.50–2.46 (m, 8H), 2.25 (s, 3H). ^13^C NMR (101 MHz, chloroform-*d*) δ 191.68, 145.69, 135.34, 129.59(2C), 129.28(2C), 62.44, 54.95(2C), 53.03(2C), 45.91. HRMS (ESI) *m*/*z* calcd for C_13_H_19_N_2_O^+^ (M + H)^+^ 219.14919, found 219.14938.

(3) *4-((4-methoxypiperidin-1-yl)methyl)benzaldehyde* (**5c**), a yellow oil, yield 45.51%. ^1^H NMR (400 MHz, chloroform-*d*) δ 9.98 (s, 1H), 7.83 (d, *J* = 8.1 Hz, 2H), 7.50 (d, *J* = 7.9 Hz, 2H), 3.56 (s, 2H), 3.33 (s, 3H), 3.23 (tq, *J* = 8.8, 4.9, 4.4 Hz, 1H), 2.72 (dq, *J* = 10.9, 5.6, 4.4 Hz, 2H), 2.17 (ddt, *J* = 18.5, 11.9, 5.9 Hz, 2H), 1.95–1.85 (m, 2H), 1.61 (ddt, *J* = 13.6, 9.4, 4.7 Hz, 2H). ^13^C NMR (101 MHz, chloroform-*d*) δ 191.96, 146.22, 135.39, 129.73(2C), 129.34(2C), 76.15, 62.61, 55.49, 51.09(2C), 30.83(2C). HRMS (ESI) *m*/*z* calcd for C_14_H_20_NO_2_^+^ (M + H)^+^ 234.14886, found 234.14902.

(4) *4-((4-methylpiperidin-1-yl)methyl)benzaldehyde* (**5d**), a yellow oil, yield 47.04%. ^1^H NMR (400 MHz, chloroform-*d*) δ 9.98 (s, 1H), 7.82 (d, *J* = 7.9 Hz, 2H), 7.49 (d, *J* = 7.9 Hz, 2H), 3.54 (s, 2H), 2.83 (ddt, *J* = 15.2, 12.5, 3.5 Hz, 2H), 1.97 (td, *J* = 11.5, 2.5 Hz, 2H), 1.59 (dt, *J* = 7.9, 5.2 Hz, 2H), 1.64–1.54 (m, 3H), 0.92 (d, *J* = 6.1 Hz, 3H). ^13^C NMR (101 MHz, chloroform-*d*) δ 191.99, 146.50, 135.34, 129.69(2C), 129.43(2C), 63.13, 54.08(2C), 34.33(2C), 30.67, 21.90. HRMS (ESI) *m*/*z* calcd for C_14_H_20_NO^+^ (M + H)^+^ 218.15394, found 218.15402.

(5) *4-(morpholinomethyl)benzaldehyde* (**5e**). A yellow solid, yield 63.00%, m.p. 40.1–40.8 °C; ^1^H NMR (400 MHz, chloroform-*d*) δ 9.99 (s, 1H), 7.84 (d, *J* = 7.9 Hz, 2H), 7.52 (d, *J* = 7.9 Hz, 2H), 3.72 (q, *J* = 4.5 Hz, 4H), 3.58 (s, 2H), 2.46 (t, *J* = 4.8 Hz, 4H). ^13^C NMR (101 MHz, chloroform-*d*) δ 191.94, 145.28, 135.58, 129.79(2C), 129.51(2C), 66.92(2C), 63.01, 53.65(2C). HRMS (ESI) *m*/*z* calcd for C_12_H_16_NO_2_^+^ (M + H)^+^ 206.11756, found 206.11769.

(6) *4-(pyrrolidin-1-ylmethyl)benzaldehyde* (**5f**), a yellow oil, yield 61.81%. ^1^H NMR (400 MHz, chloroform-*d*) δ 9.98 (s, 1H), 7.82 (d, *J* = 7.8 Hz, 2H), 7.50 (d, *J* = 7.9 Hz, 2H), 3.68 (s, 2H), 2.58–2.44 (m, 4H), 1.79 (td, *J* = 4.2, 2.7 Hz, 4H). ^13^C NMR (101 MHz, chloroform-*d*) δ 191.91, 146.85, 135.28, 129.72(2C), 129.17(2C), 60.32, 54.19(2C), 23.49(2C). HRMS (ESI) *m*/*z* calcd for C_12_H_16_NO^+^ (M + H)^+^ 190.12264, found 190.12271.

#### 3.2.6. Synthesis of Target Compounds

In this experiment, intermediate **4a** (0.559 g, 2 mmol) and intermediate **5a** (1.32 g, 6 mmol) were combined in a 100 mL three-neck flask with 20 mL of absolute ethanol to facilitate dissolution. Subsequently, a 10 mL solution of 20% potassium hydroxide was added to the reaction mixture, which was then heated and stirred at 60 °C for 8 h. The reaction progress was monitored using thin-layer chromatography (TLC). Upon completion of the reaction, the anhydrous ethyl alcohol was evaporated, and the remaining residue was dissolved in ethyl acetate. The ethyl acetate layer was then washed thrice with a saturated NaCl solution. The target compound’s crude product was obtained through the evaporation of ethyl acetate, followed by its separation via silica gel column chromatography purification using a mobile phase of ethyl acetate and methanol in a 5:1 ratio. The process was monitored using TLC, and the collected product fractions were spun off, added with a small amount of anhydrous ethanol, and sonicated. No crystals were precipitated. The column chromatography was repeated once, and the fractions were collected, monitored using TLC as a single spot, dissolved in anhydrous ethanol, transferred to a 10 mL beaker, and evaporated dry to yield 0.42 g of yellow oil, specifically compound **6a**. The synthesis methods for compound **6b**–**6u** were the same as those of **6a**.

(1) *(E)-1-(2-hydroxy-5-((4-hydroxypiperidin-1-yl)methyl)-4-methoxyphenyl)-3-(4-((4-hydroxypiperidin-1-yl)methyl)phenyl)prop-2-en-1-one* (**6a**). A yellow oil, yield 31.59%, ^1^H NMR (600 MHz, DMSO-d_6_) δ 10.14 (s, 1H), 8.06 (s, 1H), 7.94–7.69 (m, 2H), 7.48 (d, *J* = 7.9 Hz, 2H), 7.33 (d, *J* = 7.9 Hz, 2H), 6.64 (s, 1H), 5.59 (dd, *J* = 13.1, 2.9 Hz, 2H), 3.81 (s, 3H), 3.43 (d, *J* = 1.1 Hz, 2H), 3.36 (d, *J* = 2.2 Hz, 2H), 3.18–3.15 (m, 1H), 2.74–2.70 (m, 1H), 2.65 (ddd, *J* = 8.5, 7.4, 5.9 Hz, 4H), 2.03 (ddd, *J* = 8.7, 6.0, 4.9 Hz, 4H), 1.72–1.66 (m, 4H), 1.38 (qd, *J* = 12.7, 3.4 Hz, 4H). ^13^C NMR (151 MHz, DMSO-d_6_) δ 190.64, 164.25, 162.62, 139.61, 137.92, 129.61, 129.25(2C), 127.88, 126.93(2C), 121.22, 116.62, 114.00, 99.75, 66.75(2C), 62.23, 56.54, 55.19, 51.33(4C), 34.92(2C), 34.84(2C). HRMS (ESI) *m*/*z* calcd for C_28_H_37_N_2_O_5_^+^ (M + H)^+^ 481.26970, found 481.26999.

(2) *(E)-1-(2-hydroxy-5-((4-hydroxypiperidin-1-yl)methyl)-4-methoxyphenyl)-3-(4-((4-methoxypiperidin-1-yl)methyl)phenyl)prop-2-en-1-one* (**6b**), a yellow oil, yield 38.96%, ^1^H NMR (600 MHz, DMSO-d_6_) δ 12.14 (s, 1H), 8.07 (s, 1H), 7.91 (d, *J* = 15.4 Hz, 1H), 7.86–7.77 (m, 2H), 7.47 (d, *J* = 7.9 Hz, 1H), 7.37–7.32 (m, 2H), 6.55 (s, 1H), 5.58 (d, *J* = 4.9 Hz, 1H), 3.84 (s, 3H), 3.50–3.40 (m, 5H), 3.20 (s, 3H), 3.17–3.09 (m, 1H),2.69 (ddd, *J* = 8.2, 7.3, 5.8 Hz, 4H), 2.07 (t, *J* = 12.5 Hz, 4H), 1.83–1.76 (m, 2H), 1.73–1.65 (m, 2H), 1.45–1.34 (m, 4H). ^13^C NMR (151 MHz, DMSO-d_6_) δ 192.12, 165.46, 164.94, 144.23, 139.43, 133.68, 133.24, 129.60(2C), 129.24(2C), 121.28, 118.39, 113.55, 99.77, 76.09, 66.76, 62.13, 56.46, 55.41, 55.20, 51.33(2C), 50.97(2C), 34.90, 34.82, 31.06(2C). HRMS (ESI) *m*/*z* calcd for C_29_H_39_N_2_O_5_^+^ (M + H)^+^ 495.28535, found 495.28564.

(3) *(E)-1-(2-hydroxy-5-((4-hydroxypiperidin-1-yl)methyl)-4-methoxyphenyl)-3-(4-(pyrrolidin-1-ylmethyl)phenyl)prop-2-en-1-one* (**6c**). A yellow oil, yield 49.23%, ^1^H NMR (600 MHz, DMSO-*d*_6_) δ 13.49 (s, 1H), 8.12–8.09(m, 1H), 7.91 (d, *J* = 15.2 Hz, 1H), 7.82–7.79 (m, 1H), 7.38 (d, *J* = 7.7 Hz, 2H), 7.27–7.09 (m, 2H), 6.55 (s, 1H), 4.62 (d, *J* = 4.9 Hz, 1H), 3.84 (s, 3H), 3.68–3.58 (m, 3H), 3.40–3.32 (m, 2H), 2.73–2.66 (m, 2H), 2.39 (d, *J* = 17.5 Hz, 4H), 2.22 (t, *J* = 11.0 Hz, 2H), 1.82–1.76 (m, 2H), 1.66 (d, *J* = 11.5 Hz, 4H), 1.55 (t, *J* = 10.5 Hz, 2H). ^13^C NMR (151 MHz, DMSO-*d*_6_) δ 192.21, 165.60, 164.98, 144.40, 139.74, 134.99, 133.35, 129.42, 129.33, 128.90, 128.39, 121.05, 118.03, 113.44, 99.79, 70.40, 65.75, 56.43, 55.39, 53.94(2C), 50.48(2C), 30.96(2C), 23.58(2C).HRMS (ESI) *m*/*z* calcd for C_27_H_35_N_2_O_4_^+^ (M + H)^+^ 451.25913, found 451.25815.

(4) *(E)-1-(2-hydroxy-5-((4-hydroxypiperidin-1-yl)methyl)-4-methoxyphenyl)-3-(4-((4-methylpiperazin-1-yl)methyl)phenyl)prop-2-en-1-one* (**6d**). A yellow oil, yield 52.31%, ^1^H NMR (600 MHz, DMSO-d_6_) δ 10.03 (s, 1H), 8.09 (s, 1H), 7.95 (d, *J* = 15.4 Hz, 1H), 7.85–7.82 (m, 1H), 7.24–7.29 (m, 2H), 7.32–7.19 (m, 2H), 6.66(s, 1H), 4.46 (s, 1H), 3.82 (d, *J* = 3.0 Hz, 3H), 3.44–3.40 (m, 5H), 2.74–7.71 (m, 2H), 2.21–2.32 (m, 8H), 2.13 (s, 3H), 2.04–1.95 (m, 2H), 1.83–1.77 (m, 2H), 1.55 (ddd, *J* = 8.8, 5.9, 4.9 Hz, 2H). ^13^C NMR (151 MHz, DMSO-d_6_) δ 196.08, 163.85, 162.66, 141.58, 137.03, 129.33, 129.28, 129.02(2C), 128.42, 126.95, 126.76(2C), 116.62, 99.76, 66.74, 63.16, 62.34, 62.17, 55.17(4C), 52.99, 52.96, 46.19(2C), 31.01. HRMS (ESI) *m*/*z* calcd for C_28_H_38_N_3_O_4_^+^ (M + H)^+^ 480.28568, found 480.28543.

(5) (*E)-1-(2-hydroxy-5-((4-hydroxypiperidin-1-yl)methyl)-4-methoxyphenyl)-3-(4-(morpholinomethyl)phenyl)prop-2-en-1-one* (**6e**). A red oil, yield 62.35%, ^1^H NMR (600 MHz, DMSO-*d*_6_) δ 12.63 (s, 1H), 7.98 (s, 1H), 7.73–7.70 (m, 2H), 7.50–7.45 (m, 2H), 7.35 (d, *J* = 7.9 Hz, 2H), 6.49 (s, 1H), 4.50 (d, *J* = 4.9 Hz, 1H), 3.82 (s, 3H), 3.58–3.44 (m, 7H), 3.38 (s, 2H), 2.70 (dd, *J* = 8.7, 5.9 Hz, 2H), 2.55–2.51 (m, 4H), 2.11–2.08 (m, 2H), 1.71 (dt, *J* = 12.3, 3.9 Hz, 2H), 1.41 (dtd, *J* = 12.7, 9.6, 3.5 Hz, 2H). ^13^C NMR (151 MHz, DMSO-*d*_6_) δ 203.36, 164.40, 163.76, 144.90, 138.13, 133.15(4C), 129.44, 126.92, 118.06,116.62, 113.33, 99.44, 66.63(2C), 62.53, 56.51, 56.47, 56.38, 55.08(2C), 51.13(2C), 34.70(2C). HRMS (ESI) *m*/*z* calcd for C_27_H_35_N_2_O_5_^+^ (M + H)^+^ 467.25405, found 467.25305.

(6) *(E)-1-(2-hydroxy-5-((4-hydroxypiperidin-1-yl)methyl)-4-methoxyphenyl)-3-(4-((4-methylpiperidin-1-yl)methyl)phenyl)prop-2-en-1-one* (**6f**). A red oil, yield 54.70%, ^1^H NMR (600 MHz, DMSO-d_6_) δ 13.47 (s, 1H), 8.08 (s, 1H), 7.89 (d, *J* = 16.1 Hz, 1H), 7.84–7.79 (m, 2H), 7.71 (d, *J* = 15.6 Hz, 1H), 7.37–7.34 (m, 2H), 6.55 (s, 1H), 4.55 (s, 1H), 3.84 (s, 3H), 3.45–3.40 (m, 3H), 3.38 (s, 2H), 2.74 (ddd, *J* = 7.1, 5.6, 2.1 Hz, 4H), 2.06 (dd, *J* = 8.7, 5.9 Hz, 2H), 1.89 (dd, *J* = 8.1, 5.4 Hz, 2H), 1.70 (dq, *J* = 13.4, 4.0 Hz, 2H), 1.53 (d, *J* = 12.8 Hz, 2H), 1.41 (d, *J* = 10.1 Hz, 2H), 1.31–1.26 (m, 1H), 1.15–1.08 (m, 2H), 1.05 (t, *J* = 6.3 Hz, 3H). ^13^C NMR (151 MHz, DMSO-d_6_) δ 191.71, 164.49, 163.91, 143.88, 142.03, 133.12, 132.73, 129.11, 128.86, 128.74, 128.42, 120.60, 118.06, 112.95, 99.27, 66.23, 62.10, 55.96, 55.89, 53.28, 53.24, 50.60(2C), 34.38(2C), 33.93, 33.90, 30.26, 21.76. HRMS (ESI) *m*/*z* calcd for C_29_H_39_N_2_O_4_^+^ (M + H)^+^ 479.29043, found 479.29092.

(7) *(E)-1-(2-hydroxy-4-methoxy-5-((4-methoxypiperidin-1-yl)methyl)phenyl)-3-(4-((4-methylpiperidin-1-yl)methyl)phenyl)prop-2-en-1-one* (**6g**). A red oil, yield 63.48%, ^1^H NMR (600 MHz, DMSO-d_6_) δ 13.48 (s, 1H), 8.10 (s, 1H), 7.92 (d, J =15.6, 1H), 7.85–7.81 (m, 1H), 7.31–7.15 (m, 4H), 6.56 (s, 1H), 4.47 (s, 3H), 3.87–3.80 (m, 1H), 3.48–3.44 (m, 2H), 3.40 (s, 3H), 3.20 (d, *J* = 8.0 Hz, 2H), 2.76–2.71 (m, 4H), 2.12– 2.06 (m, 2H), 1.91–1.89 (qd, *J* = 12.4, 11.6, 3.0 Hz, 2H), 1.87–1.78 (m, 2H), 1.56–1.50 (m, 4H), 1.33–1.27 (m, 1H), 1.10 (qt, *J* = 13.2, 6.6 Hz, 2H), 0.87 (d, *J* = 6.5 Hz, 3H). ^13^C NMR (151 MHz, DMSO-d_6_) δ 191.76, 165.08, 164.51, 143.91, 136.88, 132.80, 129.11, 128.47(2C), 126.23(2C), 120.65, 117.85, 112.98, 99.31, 75.65, 62.24, 55.97, 54.91, 54.71, 53.24(2C), 53.17(2C), 33.90(2C), 30.56, 30.27(2C), 21.76. HRMS (ESI) m/z calcd for C_30_H_41_N_2_O_4_^+^ (M + H)^+^ 493.30608, found 493.30673.

(8) *(E)-1-(2-hydroxy-4-methoxy-5-((4-methoxypiperidin-1-yl)methyl)phenyl)-3-(4-(morpholinomethyl)phenyl)prop-2-en-1-one* (**6h**). A red oil, yield 63.56%, ^1^H NMR (600 MHz, DMSO-*d*_6_) δ 12.62 (s, 1H), 7.91–7.71 (m, 3H), 7.50–7.48 (m, 2H), 7.37–7.35 (m, 2H),6.52 (s, 1H), 3.87–3.83 (m, 4H), 3.65–3.51 (m, 4H), 3.47–3.41 (m, 4H), 3.21 (s, 3H), 2.74–2.69 (m, 2H), 2.56 (d, *J* = 2.4 Hz, 4H), 2.34–2.21 (m, 2H), 1.84 (d, *J* = 12.8 Hz, 2H), 1.46 (s, 2H). ^13^C NMR (151 MHz, DMSO-*d*_6_) δ 202.91, 163.95, 163.42, 140.92, 137.67, 133.21, 131.17, 128.95(2C), 126.45(2C), 112.91, 117.53, 112.91, 99.03, 75.14, 66.14(2C), 62.04, 55.96, 54.77, 54.37, 53.11(2C), 50.10(2C), 30.20(2C). HRMS (ESI) *m*/*z* calcd for C_28_H_37_N_2_O_5_^+^ (M + H)^+^ 481.26970, found 481.27039.

(9) *(E)-1-(2-hydroxy-4-methoxy-5-(pyrrolidin-1-ylmethyl)phenyl)-3-(4-((4-hydroxypiperidin-1-yl)methyl)phenyl)prop-2-en-1-one* (**6i**). A yellow oil, yield 65.37%. ^1^H NMR (600 MHz, DMSO-*d*_6_) δ 13.45 (s, 1H), 8.11 (s, 1H), 7.93 (d, *J* = 15.4 Hz, 1H), 7.86–7.80 (m, 2H), 7.50–7.46 (m, 1H), 7.38 (d, *J* = 7.9 Hz, 2H), 6.55 (s, 1H), 4.55 (d, *J* = 4.9 Hz, 1H), 3.86 (s, 3H), 3.56 (s, 2H), 3.53–3.51 (m, 1H), 3.47 (s, 2H), 2.69–2.60 (m, 4H), 2.46 (q, *J* = 7.6, 6.6 Hz, 4H), 2.01 (dt, *J* = 29.6, 11.0 Hz, 2H), 1.73–1.64 (m, 4H), 1.39 (ddt, *J* = 22.1, 12.4, 6.3 Hz, 2H). ^13^C NMR (151 MHz, DMSO-*d*_6_) δ 191.76, 164.27, 163.50, 143.91, 137.16, 133.15, 132.53, 129.09, 128.94, 128.74, 128.44, 120.71,116.13, 112.95, 99.26, 66.28, 61.89, 56.03, 55.95, 53.46, 52.42, 50.85(2C), 34.37(2C), 22.99(2C). HRMS (ESI) *m*/*z* calcd for C_27_H_35_N_2_O_4_^+^ (M + H)^+^ 451.25913, found 451.25922.

(10) *(E)-1-(2-hydroxy-4-methoxy-5-(pyrrolidin-1-ylmethyl)phenyl)-3-(4-(morpholinomethyl)phenyl)prop-2-en-1-one* (**6j**). A red oil, yield 68.30%. ^1^H NMR (600 MHz, DMSO-*d*_6_) δ 13.54 (s, 1H), 7.85–7.11 (m, 3H), 7.39 (d, *J* = 7.8 Hz, 2H), 7.35 (d, *J* = 8.1 Hz, 2H), 6.49 (d, 1H), 3.82 (s, 3H), 3.60–3.53 (m, 4H), 3.49 (d, *J* = 6.8 Hz, 2H), 3.46 (d, *J* = 4.1 Hz, 2H),2.56–2.53 (m, 2H), 2.47–2.40 (m, 4H), 2.37–2.32 (m, 2H), 1.67 (dt, *J* = 4.5, 3.1 Hz, 4H). ^13^C NMR (151 MHz, DMSO-*d*_6_) δ 202.86, 163.67, 163.23, 143.75, 140.98, 132.38, 132.29, 129.27(2C), 128.91(2C), 120.80, 118.86, 112.80, 98.88, 66.15(2C), 62.05, 55.97, 55.83, 53.35(2C), 52.39(2C), 23.08(2C). HRMS (ESI) *m*/*z* calcd for C_26_H_33_N_2_O_4_^+^ (M + H)^+^ 437.24348, found 437.24399.

(11) *(E)-1-(2-hydroxy-4-methoxy-5-(pyrrolidin-1-ylmethyl)phenyl)-3-(4-((4-methylpiperidin-1-yl)methyl)phenyl)prop-2-en-1-one* (**6k**). A red oil, yield 65.66%, ^1^H NMR (600 MHz, DMSO-*d*_6_) δ 13.48 (s, 1H), 8.08 (s, 1H), 7.90 (d, *J* = 15.4 Hz, 1H), 7.83–7.74 (m, 2H), 7.36 (dt, *J* = 7.9 Hz, 1.0 Hz, 2H), 7.31 (d, *J* = 8.1 Hz, 1H), 6.54 (s, 1H), 3.85 (s, 3H), 3.55 (s, 2H), 3.45 (s, 2H), 2.76–2.69 (m, 2H), 2.48–2.41 (m, 4H), 1.88 (qd, *J* = 11.0, 10.4, 2.5 Hz, 2H), 1.70–1.63 (m, 4H), 1.53 (d, *J* = 12.8 Hz, 2H), 1.29 (dtt, *J* = 15.0, 7.4, 3.7 Hz, 1H), 1.12 (tt, *J* = 11.8, 5.7 Hz, 2H), 0.86 (d, *J* = 6.5 Hz, 3H). ^13^C NMR (151 MHz, DMSO-*d*_6_) δ 191.74, 165.01, 164.24, 143.85, 142.01, 133.13, 132.36, 129.07(2C), 128.85(2C), 120.66, 119.00, 112.97, 99.22, 62.11, 55.99, 55.90, 53.45, 53.27, 53.24, 53.17, 33.93(2C), 30.21, 23.10, 23.02, 21.72. HRMS (ESI) *m*/*z* calcd for C_28_H_37_N_2_O_3_^+^ (M + H)^+^ 449.27987, found 449.28000.

(12) *(E)-1-(2-hydroxy-4-methoxy-5-(pyrrolidin-1-ylmethyl)phenyl)-3-(4-((4-methoxypiperidin-1-yl)methyl)phenyl)prop-2-en-1-one* (**6l**). A yellow oil, yield 54.79%, ^1^H NMR (600 MHz, DMSO-*d*_6_) δ 12.61 (s, 1H), 8.11 (s, 1H), 7.83 –7.72 (m, 2H), 7.47–7.42 (m, 2H), 7.33 (dt, *J* = 8.6, 1.1 Hz, 2H), 6.49 (s, 1H), 3.83 (s, 3H), 3.49 (s, 3H), 3.38 (s, 2H), 3.20 (s, 2H), 3.13 (dq, *J* = 8.9, 4.4 Hz, 1H), 2.63–2.58 (m, 2H), 2.46–2.44 (m, 4H), 2.06–1.99 (m, 2H), 1.81–1.76 (m, 2H), 1.68 (h, *J* = 2.7 Hz, 4H), 1.43–1.35 (m, 2H). ^13^C NMR (151 MHz, DMSO-*d*_6_) δ 202.86, 163.67, 163.22, 141.39, 137.08, 132.32, 132.30, 128.42(4C), 123.16, 118.84, 112.82, 98.91, 75.70, 61.82, 55.85, 55.84, 54.68, 53.35(2C), 52.40, 52.38, 30.60(2C), 23.08(2C). HRMS (ESI) *m*/*z* calcd for C_28_H_37_N_2_O_4_^+^ (M + H)^+^ 465.27478, found 465.27451.

(13) *(E)-1-(2-hydroxy-4-methoxy-5-(pyrrolidin-1-ylmethyl)phenyl)-3-(4-((4-methylpiperazin-1-yl)methyl)phenyl)prop-2-en-1-one* (**6m**). A red oil, yield 50.95%, ^1^H NMR (600 MHz, DMSO-*d*_6_) δ 13.44(s,1H), 8.09 (s, 1H), 7.91 (d, *J* = 16.0 Hz, 1H), 7.85–7.76 (m, 1H), 7.48 (d, *J* = 8.0 Hz, 2H), 7.33 (d, *J* = 7.8 Hz, 2H), 6.63 (s, 1H), 3.82 (s, 3H), 3.55 (s, 2H), 3.50 (s, 2H), 2.73 (dd, *J* = 16.7, 3.0 Hz, 4H), 2.48–2.41 (m, 8H), 2.14 (s, 3H), 1.68 (qq, *J* = 6.8, 3.7, 3.0 Hz, 4H). ^13^C NMR (151 MHz, DMSO-*d*_6_) δ 190.03, 163.51, 162.11, 143.79, 138.68, 133.26, 132.41, 128.82(2C), 126.41(2C), 120.84, 119.04, 113.52, 99.19, 61.68, 56.00, 55.92, 54.68(2C), 53.46(2C), 52.51(2C), 45.68, 23.11(2C). HRMS (ESI) *m*/*z* calcd for C_27_H_36_N_3_O_3_^+^ (M + H)^+^ 450.27512, found 450.27533.

(14) *(E)-1-(2-hydroxy-4-methoxy-5-(morpholinomethyl)phenyl)-3-(4-(morpholinomethyl)phenyl)prop-2-en-1-one* (**6n**). A red oil, yield 61.07%, ^1^H NMR (600 MHz, DMSO-*d*_6_) δ 12.63 (s, 1H), 8.09 (s, 1H), 7.90 (d, *J* = 15.5 Hz, 1H), 7.84–7.77 (m, 1H), 7.45–7.37 (m, 2H), 7.31–7.21 (m, 2H), 6.50 (s, 1H), 3.82 (s, 3H), 3.69–3.54 (m, 8H), 3.50–3.42 (m, 2H), 3.48–3.38 (m, 2H), 2.54–2.50 (m, 2H), 2.37 (tt, *J* = 15.8, 4.6 Hz, 4H), 2.33–2.29 (m, 2H). ^13^C NMR (151 MHz, DMSO-*d*_6_) δ 191.71, 164.02, 163.40, 143.77, 141.21, 133.33, 132.83, 128.61(2C), 126.27(2C), 120.75, 117.11, 112.85, 99.01, 66.19(2C), 66.15(2C), 62.73, 55.86, 55.21, 53.14, 53.08, 53.03, 52.96. HRMS (ESI) *m*/*z* calcd for C_26_H_33_N_2_O_5_^+^ (M + H)^+^ 453.23840, found 453.23837.

(15) *(E)-1-(2-hydroxy-4-methoxy-5-(morpholinomethyl)phenyl)-3-(4-((4-methylpiperidin-1-yl)methyl)phenyl)prop-2-en-1-one* (**6o**). A yellow solid, yield 53.22%, m.p. 104.2–105.4 °C, ^1^H NMR (600 MHz, chloroform-*d*) δ 13.50 (s, 1H), 7.88 (d, *J* = 15.6 Hz, 2H), 7.63–7.56 (m, 3H), 7.40 (d, *J* = 8.1 Hz, 2H), 6.45 (s, 1H), 3.86 (s, 3H), 3.75–3.71 (m, 6H), 3.51 (d, *J* = 9.2 Hz, 2H), 2.84 (dt, *J* = 11.8, 3.3 Hz, 2H), 2.52 (t, *J* = 4.6 Hz, 4H), 2.00–1.93 (m, 2H), 1.63–1.57 (m, 2H), 1.36 (ddt, *J* = 13.7, 9.6, 4.1 Hz, 1H), 1.26 (qd, *J* = 11.9, 3.6 Hz, 2H), 0.92 (d, *J* = 6.2 Hz, 3H). ^13^C NMR (151 MHz, DMSO-*d*_6_) δ 186.66, 160.73, 159.29, 139.04, 136.89, 128.24, 126.39, 124.45(2C), 123.22(2C), 114.60, 112.21, 108.25, 94.23, 61.83(2C), 57.91, 50.66, 50.55, 48.79(2C), 48.30(2C), 29.09(2C), 25.48, 21.06. HRMS (ESI) *m*/*z* calcd for C_28_H_37_N_2_O_4_^+^ (M + H)^+^ 465.27478, found 465.27505.

(16) (*E)-1-(2-hydroxy-4-methoxy-5-(morpholinomethyl)phenyl)-3-(4-((4-methoxypiperidin-1-yl)methyl)phenyl)prop-2-en-1-one* (**6p**). A yellow solid, yield 62.04%, m.p. 139.1–140.2 °C, ^1^H NMR (600 MHz, chloroform-*d*) δ 13.48 (s, 1H), 7.86 (d, *J* = 13.6 Hz, 2H), 7.62–7.55 (m, 3H), 7.39 (d, *J* = 7.8 Hz, 2H), 6.44 (s, 1H), 3.85 (s, 3H), 3.73 (t, *J* = 4.6 Hz, 4H), 3.51 (s, 2H), 3.48 (s, 2H), 3.32 (s, 3H), 3.22 (td, *J* = 8.7, 4.2 Hz, 1H), 2.71 (d, *J* = 12.4 Hz, 2H), 2.50 (t, *J* = 4.7 Hz, 4H), 2.16 (t, *J* = 10.8 Hz, 2H), 1.89 (dt, *J* = 14.2, 4.0 Hz, 2H), 1.60 (qd, *J* = 12.3, 10.7, 3.4 Hz, 2H). ^13^C NMR (151 MHz, DMSO-*d*_6_) δ 186.62, 160.73, 159.30, 138.96, 136.70, 128.33, 126.41, 124.31(2C), 123.26(2C), 114.67, 112.21, 108.23, 94.21, 71.06, 61.81(2C), 57.40, 50.64, 50.54, 50.27, 48.28(2C), 45.84(2C), 25.64(2C). HRMS (ESI) *m*/*z* calcd for C_28_H_37_N_2_O_5_^+^ (M + H)^+^ 481.26970, found 481.26999.

(17) *(E)-1-(2-hydroxy-4-methoxy-5-((4-methylpiperidin-1-yl)methyl)phenyl)-3-(4-(morpholinomethyl)phenyl)prop-2-en-1-one* (**6q**). A yellow oil, yield 71.22%, ^1^H NMR (600 MHz, DMSO-*d*_6_) δ 12.60 (s, 1H), 8.07 (s, 1H), 7.91–7.71 (m, 2H), 7.26–7.25 (m, 2H), 7.24–7.20 (m, 2H), 6.49 (s, 1H), 3.82 (s, 3H), 3.56 (dt, *J* = 14.3, 4.5 Hz, 4H), 3.47–3.37 (m, 2H), 3.36 (s, 2H), 2.78 (dt, *J* = 12.4, 3.3 Hz, 2H), 2.38–2.30 (m, 4H), 1.94 (qd, *J* = 14.2, 11.7, 7.1 Hz, 2H), 1.57–1.51 (m, 2H), 1.29 (tdd, *J* = 12.2, 6.5, 3.3 Hz, 1H), 1.13 (qd, *J* = 11.9, 3.8 Hz, 2H), 0.87 (d, *J* = 6.6 Hz, 3H). ^13^C NMR (151 MHz, DMSO-*d*_6_) δ 202.81, 163.94, 163.20, 141.20, 136.01, 132.39, 128.68, 128.61(2C), 126.27(2C), 120.84, 118.00, 112.85, 98.93, 66.15(2C), 62.71, 55.86, 55.07, 53.19(2C), 53.09(2C), 33.98(2C), 30.24, 21.73. HRMS (ESI) *m*/*z* calcd for C_28_H_37_N_2_O_4_^+^ (M + H)^+^ 465.27478, found 465.27518.

(18) *(E)-1-(2-hydroxy-4-methoxy-5-((4-methylpiperidin-1-yl)methyl)phenyl)-3-(4-((4-methylpiperidin-1-yl)methyl)phenyl)prop-2-en-1-one* (**6r**). A red oil, yield 58.74%. ^1^H NMR (600 MHz, DMSO-d_6_) δ 12.60 (s, 1H), 8.07 (s, 1H), 7.91–7.70 (m, 2H), 7.23–7.28 (m, 2H), 7.22–7.19 (m, 2H), 6.50 (s, 1H), 3.82 (s, 3H), 3.39 (s, 2H), 3.35 (s, 2H), 2.76 (ddq, *J* = 29.8, 12.0, 4.1, 3.5 Hz, 4H), 1.99–1.90 (m, 2H), 1.86 (td, *J* = 11.7, 2.5 Hz, 2H), 1.58–1.50 (m, 4H), 1.35–1.25 (m, 2H), 1.18–1.03 (m, 4H), 0.87 (t, *J* = 6.4 Hz, 6H). ^13^C NMR (151 MHz, DMSO-d_6_) δ 202.86, 163.95, 163.18, 140.96, 136.95, 132.42, 129.10, 128.45(2C), 126.23(2C), 120.61, 118.01, 112.85, 98.95, 62.74, 55.89, 55.09, 53.27(2C), 53.19(2C), 33.99(2C), 33.92(2C), 30.28, 30.25, 21.76(2C). HRMS (ESI) *m*/*z* calcd for C_30_H_41_N_2_O_3_^+^ (M + H)^+^ 477.31117, found 477.31140.

(19) *(E)-1-(2-hydroxy-4-methoxy-5-((4-methylpiperidin-1-yl)methyl)phenyl)-3-(4-(pyrrolidin-1-ylmethyl)phenyl)prop-2-en-1-one* (**6s**). A red oil, yield 75.38%. ^1^H NMR (600 MHz, DMSO-*d*_6_) δ 13.44 (s, 1H), 8.08 (s, 1H), 7.93–7.84 (m, 2H), 7.40–7.33 (m, 2H), 7.24 (dt, *J* = 8.6, 1.1 Hz, 2H), 6.55 (s, 1H), 3.84 (s, 3H), 3.60 (d, *J* = 1.1 Hz, 2H), 3.52 (d, *J* = 2.1 Hz, 2H), 2.78 (dq, *J* = 10.8, 3.5 Hz, 2H), 2.41 (dt, *J* = 19.4, 5.9 Hz, 4H), 1.98–1.88 (m, 2H), 1.67 (dtt, *J* = 10.4, 6.6, 3.4 Hz, 4H), 1.54 (dt, *J* = 13.0, 4.3 Hz, 2H), 1.34–1.25 (m, 1H), 1.18–1.04 (m, 2H), 0.86 (dd, *J* = 12.7, 6.5 Hz, 3H). ^13^C NMR (151 MHz, DMSO-*d*_6_) δ 191.75, 164.51, 163.77, 140.90, 137.70, 133.07, 128.87, 128.12(2C), 126.23(2C), 120.66, 118.07, 112.96, 99.25, 62.73, 59.36, 59.28, 53.45, 53.38(2C), 53.03, 33.99(2C), 30.26, 23.06(2C), 21.77. HRMS (ESI) *m*/*z* calcd for C_28_H_37_N_2_O_3_^+^ (M + H)^+^ 449.27987, found 449.28003.

(20) *(E)-1-(2-hydroxy-4-methoxy-5-((4-methylpiperidin-1-yl)methyl)phenyl)-3-(4-((4-methylpiperazin-1-yl)methyl)phenyl)prop-2-en-1-one* (**6t**). A red oil, yield 79.18% ^1^H NMR (600 MHz, DMSO-d_6_) δ 13.24 (s, 1H), 8.06 (s, 1H), 7.96–7.77 (m, 2H), 7.48 (d, *J* = 7.9 Hz, 2H), 7.33 (d, *J* = 7.7 Hz, 2H), 6.63 (s, 1H), 3.81 (s, 3H), 3.45 (d, *J* = 1.1 Hz, 2H), 3.36 (d, *J* = 2.1 Hz, 2H), 2.81–2.69 (m, 2H), 2.54–2.34 (m, 8H), 1.90 (dd, *J* = 11.7, 2.5 Hz, 2H), 1.77 (s, 3H), 1.57–1.52 (m, 2H), 1.33–1.26 (m, 1H), 1.13 (qd, *J* = 12.5, 4.2 Hz, 2H), 0.92–0.82 (m, 3H). ^13^C NMR (151 MHz, DMSO-d_6_) δ 190.12, 163.77, 162.09, 138.66, 137.54, 129.20, 128.84(2C), 127.36, 126.44(2C), 120.77, 116.13, 113.50, 99.24, 61.67, 56.03, 55.09, 54.65(2C), 53.31(2C), 52.49(2C), 45.68, 33.98(2C), 30.25, 21.78. HRMS (ESI) *m*/*z* calcd for C_29_H_40_N_3_O_3_^+^ (M + H)^+^ 478.30642, found 478.30682.

(21) *(E)-1-(2-hydroxy-4-methoxy-5-((4-methylpiperazin-1-yl)methyl)phenyl)-3-(4-((4-methylpiperidin-1-yl)methyl)phenyl)prop-2-en-1-one* (**6u**). A red oil, yield 57.10%. ^1^H NMR (600 MHz, DMSO-d_6_) δ 13.43 (s, 1H), 8.09 (s, 1H), 7.93–7.91 (m, 1H), 7.85–7.75 (m, 2H), 7.72 (s, 1H), 7.38 (d, *J* = 8.0 Hz, 2H), 6.51 (s, 1H), 3.82 (s, 3H), 3.43 (d, *J* = 0.9 Hz, 2H), 3.37 (d, *J* = 2.2 Hz, 2H), 2.75 (dt, *J* = 12.2, 3.4 Hz, 2H), 2.41–2.31 (m, 8H), 2.13 (s, 3H), 1.91 (td, *J* = 11.7, 2.5 Hz, 2H), 1.55 (d, *J* = 12.8 Hz, 2H), 1.36–1.27 (m, 1H), 1.18–1.09 (m, 2H), 0.88 (d, *J* = 6.6 Hz, 3H). ^13^C NMR (151 MHz, DMSO-d_6_) δ 202.94, 163.96, 163.22, 143.89, 138.30, 133.12, 132.62, 129.14(2C), 128.89(2C), 120.71, 117.58, 112.93, 99.01, 62.07, 56.00, 55.94, 54.71(2C), 53.27(2C), 52.39(2C), 45.66, 33.92(2C), 30.22, 21.78. HRMS (ESI) *m*/*z* calcd for C_29_H_40_N_3_O_3_^+^ (M + H)^+^ 478.30642, found 478.30673.

### 3.3. Biological Assays

#### 3.3.1. Cell Lines and Cell Culture

HeLa and SiHa cell lines were obtained from the Procell Life Science Technology Co., Ltd. (Wuhan, China); H8 cell line was obtained from the Shanghai HonSun Biological Technology Co., Ltd. (Shanghai, China); HUVEC cell line was obtained from the Saibaikang (Shanghai) Biotechnology Co., Ltd. (Shanghai, China); HeLa/DDP cell line was obtained from the Guangzhou Jenniobio Biotechnology Co., Ltd. (Guangzhou, China). HeLa, SiHa, H8, and HeLa/DDP cells were cultured in DMEM medium (HyClone). The VEGF of 50 ng/mL was used as the promoter of the VEGFR-2 receptor on HUVEC cell membrane surfaces, and their inhibitory effects on VEGFR-2 were determined. All the mediums were supplemented with 10% FBS (HyClone, GE Healthcare, Australia). Cells were maintained at subconfluency at 37 °C in humidified air containing 5% CO_2_. The cells were monitored daily and maintained at 80% cell density. All compounds tested were dissolved in DMSO and then diluted by the culture medium before the treatment of cultured cells. Identification reports are available for all types of cell lines involved in this study.

#### 3.3.2. In Vitro Cytotoxicity Evaluation

The cytotoxicity of the target compounds were measured in vitro using the MTT assay. Cells grown in the logarithmic phase were counted and seeded in 96-well cell culture plates (5 × 10^3^ cells per well) for 24 h to allow attachment of cells to the wall of the plate. The old culture solution was replaced with 200 μL solution of different concentrations of 1.0, 6.25, 12.5, 25, 50, and 100 μM and various positive drugs and cultured for 48 h. Next, 20 μL of 5 mg.mL^−1^ MTT solution was added to each well and cultured for 4 h. The old solution was discarded, replaced with 150 μL DMSO solution per well, and shaken for 10 min. The OD value was measured under 490 nm, and the inhibition rate was calculated. The nonlinear regression fitting of the cell growth rate was carried out using SPSS 23.0 software, and the IC_50_ of the compound was calculated. The calculation formula was inhibition rate = [(OD_empty_ − OD_test_)/(OD_empty_ − OD_negative group_)] × 100%, (OD_empty_ blank group absorbance, OD_test_ absorbance of intervention group, OD_negative group_ without cell culture medium). There were 6 compound holes in each group, and the experimental results were presented as the average of 3 experiments. A complete medium containing only cells was used as the blank control, whereas a complete medium without cells was the negative control group. Cisplatin and sorafenib were utilized as the positive control group, and the target compound was the experimental intervention group.

#### 3.3.3. In Vitro Anti-HUVEC Cell Activities and Western Blot Analysis

The proliferation inhibitory activity of the lead compounds chalcone, **6f**, **6k**, and sorafenib on HUVEC cells was determined in the same way as in 3.3.2. HUVEC cell was seeded in 60 mm Petri dishes(1.0 × 10^6^ cells/dish) and incubated with 0.1% DMSO (vehicle), positive control sorafenib (4 μM for HUVEC), and various concentrations of compound **6f** (2, 4, and 6 μM for HUVEC), respectively. After incubation for 24 h, the cell was collected using centrifugation and washed twice using PBS chilled to 0 °C. Then, the cells were homogenized in RIPA lysis buffer and 1% PMSF, 1% mixed phosphatase inhibitor (Solarbio, Beijing, China). The lysates were incubated on ice for 30 min, intermittently vortexed every 5 min, and centrifuged at 12,000 rpm for 20 min to harvest the supernatants.

Next, the protein concentrations were determined using a BCA Protein Assay Kit (Solarbio, China). An equal amount of the proteins (12 μg) were separated using 8–12% sodium dodecyl sulfate–polyacrylamide gel electrophoresis (SDS-PAGE) and transferred to nitrocellulose membranes (Amersham Biosciences, Little Chalfont, Buckinghamshire, UK). Then, the membranes were blocked with 5% nonfat dried milk in TBS containing 1% Tween-20 for 2 h at room temperature and were incubated overnight with specific primary antibodies (CST, Boston, MA, USA) at 4 °C. After three washes in TBST, the membranes were incubated with the appropriate HRP-conjugated secondary antibodies at room temperature for 2 h. The blots were developed using enhanced chemiluminescence (Biosharp, Shanghai, China) and were detected using a PE 0723 imager (ProteinSimple, Silicon Valley, CA, USA). Each experiment was performed at least in triplicate and analyzed using Image J software (V1.8.0.112). VEGFR-2 (ab134191) rabbit mAb, p-VEGFR-2 (ab194806) rabbit mAb, PI3 kinase p85 (19H8) rabbit mAb #4257 (CST), phospho-PI3 kinase p85 (Tyr458)/p55 (Tyr199) (E3U1H) rabbit mAb #17366 (CST), Akt (pan) (C67E7) rabbit mAb #4691(CST), phospho-Akt (Ser473) (D9E) XP^®^ rabbit mAb #4060 (CST), MDR1/ABCB1 (E1Y7B) rabbit mAb #13342 (CST), Bax (ab289364), Bcl-2 (ab218123), β-actin(ab198991) rabbit mAb, goat anti-rabbit IgG (H + L) HRP—#S0001 (Affinity).

#### 3.3.4. VEGFR-2 Inhibition Test

The human p-VEGFR-2 enzyme-linked immunosorbent assay (ELISA, Human p-VEGFR-2 ELISA Kit, Sino Best Biological Technology Co., Ltd., Shanghai, China) was used to evaluate the in vitro VEGFR-2 inhibitory activity of compounds **6f**, **6k**, and sorafenib by detecting p-VEGFR-2 expression in HeLa cells. Specimens, standards, and HRP-labeled detection antibodies were added sequentially to the pre-covered microtiter wells containing phosphorylated vascular endothelial growth factor receptor 2 (p-VEGFR2) antibodies, warmed, and washed thoroughly. The substrate TMB was added for color development. This substrate is converted to blue using peroxidase catalysis to a final yellow color due to the action of acid. The color shade was positively correlated with the phosphorylated vascular endothelial growth factor receptor 2 (p-VEGFR2) in the sample, and the absorbance (OD) was measured at 450 nm using an enzymatic standard. Curve fitting was performed using SPSS 23.0 software to calculate IC_50_.

#### 3.3.5. Effect of Compound **6f** on PI3K/AKT Signaling Pathway

HeLa cell was seeded in 60 mm Petri dishes(1.0 × 10^6^ cells/dish) and incubated using 0.1% DMSO (vehicle), positive control sorafenib (4 μM for HeLa), and various concentrations of compound **6f** (2, 4, and 6 μM for HeLa), respectively. The procedure for the remaining Western blot was the same as in Section 3.3.3.

#### 3.3.6. Apoptosis Analysis

HeLa Cells were seeded in 60 mm Petri dishes (1.0 × 10^6^ cells/dish), incubated using 0.1% DMSO (vehicle), 4 μM of sorafenib (positive control), and different concentrations (2, 4, and 6 μM) of compound **6f** in the separately prepared medium for 24 h, respectively. After incubation, cells were harvested and incubated using 5 mL of Annexin-V/FITC (BD, America) in binding buffer (10 mM HEPES, 140 mM NaCl, and 2.5 mM CaCl_2_ at pH 7.4) at room temperature for 15 min. The PI solution was then added to the medium for another 10 min incubation. Almost 10,000 events were collected for each sample and analyzed using flow cytometry (BD, LSRFortessa, New York, NY, USA). The procedure for the remaining Western blot was the same as in Section 3.3.3.

#### 3.3.7. Transwell Migration and Invasion Assay

Initially, HeLa (5 × 10^4^ cells) were suspended in 200 μL of an FBS-free medium. The top chamber contained the vehicle and various concentrations of compound 6f (2, 4, and 6 μM) and sorafenib (4 μM). The lower chamber was filled with 600 μL of medium containing 10% FBS. After incubating at 37 °C for 24 h, the cells on the top side of the transwell membrane were removed with a cotton tip. The cells trapped on the bottom side of the membrane were fixed using methanol and stained using 0.1% crystal violet solution for 30 min, respectively.

The transwell (12 mm pore, Corning Incorporated, New York, NY, USA) was precoated using 50 μL Matrigel for 5 h at 37 °C to achieve solidification. HeLa cells were harvested and resuspended in serum-free medium containing 0, 2, 4, and 6 μM of compound **6f** compared with 4 μM of sorafenib and added into the upper wells of the transwell chamber at a density of 5 × 10^5^ cells/mL, while 600 μL of DMEM containing 10% FBS was added into the lower chambers, which had been coated using 50 μL of Matrigel (1:8 dilution in serum-free medium, Corning/BD Biosciences). After 24 h of incubation at 37 °C, the invasion cells were fixed using methanol and stained using 0.1% crystal violet for 30 min, respectively. Then, the chambers were washed using PBS and left to dry. Images were photographed using an inverted fluorescence microscope (Leica, Microsystems CMS GmbH, GER) and counted using Image J software (V1.8.0.112) for three independent fields randomly.

#### 3.3.8. Anti-Cisplatin-Resistant Cervical Cancer Activity

To reduce the adverse reactions caused by the fixation of drug resistance reversal agents, the concentration that reduces the proliferation rate of tumor cells by less than 10% was selected. Therefore, the inhibitory activity of compound **6f** on HeLa/DDP cells was determined in the concentration ranges of 0.25, 0.5, and 1 μM combined with cisplatin. The experimental method was similar to that described in Section 3.3.2. The effects of verapamil (6 μM) and **6f** (0.25, 0.5, and 1 μM) on P-gp expression were determined; the effects of sorafenib (4 μM) and **6f** (2, 4, and 6 μM) on VEGFR-2 expression in HeLa/DDP cells were further determined, the procedure for the remaining Western blot was the same as in Section 3.3.3.

#### 3.3.9. Molecular Docking Experiment

Compounds **6f** and **6k** were docked using VEGFR-2 (ID:4SAD) and P-gp (ID:7O9W) protein targets, respectively. The protein’s crystal structure was mainly obtained from the RCSB Protein Data Bank database. Small molecules were drawn into a 3D format using ChemOffice 18.0 and preserved after MM-2 energy optimization—PDB format standby. The format was transformed, and the active pocket was selected using AutodockTools1.5.6 software. Compounds **6f**, **6k**, and chalcone were used as ligands to dock with the crystal structure of VEGFR-2 and P-gp. The protein crystal removed the water molecules and ligands from the original file in PyMoL1.7.6 software, adding polar hydrogen atoms and charges. Finally, the software was used to draw a 3D map of the compound molecule binding to the protein target.

#### 3.3.10. Statistical Analysis

The results were statistically analyzed using SPSS23.0 software and expressed as (mean ± SD). If the data satisfied the variance homogeneity of normal distribution, multiple single factor analysis of variance was used, otherwise the nonparametric test was used; *p* < 0.05 was considered statistically significant.

## 4. Conclusions

In this study, a total of 21 novel chalcone derivatives were synthesized through the utilization of glycyrrhiza chalcone as the primary compound backbone and VEGFR-2 and P-gp as the action targets, employing the active substructure splicing principle. The proliferation inhibitory activity and mechanism of action of the synthesized compounds were preliminarily assessed against cervical cancer and cisplatin-resistant cervical cancer cells. The results indicate that **6f** exhibits promising therapeutic potential against cervical cancer, possibly due to its dual inhibitory effect on VEGFR-2 and P-gp activities. This study serves as a foundation for further investigations into novel molecularly targeted chalcone analogs for the treatment of cervical cancer, as well as the development of innovative antitumor multidrug-resistant drugs.

## Data Availability

The study is supported by primary data.

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
