# Peer review of "Design, Synthesis, and Anti-Cervical Cancer and Reversal of Tumor Multidrug Resistance Activity of Novel Nitrogen-Containing Heterocyclic Chalcone Derivatives"

_molecules, 2023, doi:10.3390/molecules28114537_

Round 1
Reviewer 1 Report
The authors present the synthesis of new nitrogen-containing chalcone derivatives. The cytotoxic activity of all compounds was evaluated towards two cervical cancer cell lines and normal human cervical epithelial cells. The studies also revealed a more detailed mechanism of action, including the inhibitory effect on the VEGFR-2 receptor. The paper presents interesting new data concerning the anti-cancer activity of chalcone derivatives thus, I recommend the article for publication, however, some issues have to be addressed and corrected.
1. In the abstract full name of the most active compound 6f should be included. That would make the section more clear to the reader and inform which particular structural element is associated with the elevated activity.
2. In the introduction, in the lines describing the previous studies (ref. [30] and [31]), attaching a scheme that will present the structures of the mentioned most active derivatives would improve the description of the topic.
3. Section “2.1. Synthesis” – “(Scheme 1)”, “(Scheme 2)”… should be included after each paragraph describing each scheme.
4. Section “2.1. Synthesis”, second paragraph – “TNF” is misspelled.
5. In the whole text, I would recommend changing the symbol “ Ì´”, to a typical dash “ – “.
6. Scheme 3 and 4 – The R1 and R2 attached to the nitrogen atoms of the final products is confusing. If both nitrogen atoms in the final structure (attached to ring A and B) were changed to R1 and R2 that would simplify the schemes.
7. Section “4.2.6. Synthesis of target compounds” – The sentence “Ethyl acetate was evaporated and purified with anhydrous ethanol to obtain a yellow oil” is unclear. How was the purification of the target compound exactly performed?
Reviewer 2 Report
The manuscript entitled "Design, synthesis and anti-cervical cancer and reversal of tumor multidrug resistance activity of novel nitrogen-containing heterocyclic chalcone derivatives" is a good work. However, I have some suggestions based on the manuscript;
1. Since, the article is based on the pharmacological effects of chalcones, emphasis may be given to it. The introduction may be re-written by emphasizing the role of chalcones initially.
2. The introduction must include description on the chemistry of chalcones and their pharmacological effects in detail
3. What was the reason for choosing HeLa cells?
4. I suggest to seperate results and discussion sections so that an in depth analysis of the obtained results.
5. Magnification of the microscopic images must be indicated
6. Authentication ID of cell lines may be included. Or else, authentication details/ statement must be incorporated in materials
Genereal punctuation and spelling errors must be indicated
Round 2
Reviewer 2 Report
No more comments